# Association between CHADS_2_, CHA_2_DS_2_-VASc, ATRIA, and Essen Stroke Risk Scores and Unsuccessful Recanalization after Endovascular Thrombectomy in Acute Ischemic Stroke Patients

**DOI:** 10.3390/jcm11010274

**Published:** 2022-01-05

**Authors:** Hyung Jun Kim, Moo-Seok Park, Joonsang Yoo, Young Dae Kim, Hyungjong Park, Byung Moon Kim, Oh Young Bang, Hyeon Chang Kim, Euna Han, Dong Joon Kim, JoonNyung Heo, Jin Kyo Choi, Kyung-Yul Lee, Hye Sun Lee, Dong Hoon Shin, Hye-Yeon Choi, Sung-Il Sohn, Jeong-Ho Hong, Jong Yun Lee, Jang-Hyun Baek, Gyu Sik Kim, Woo-Keun Seo, Jong-Won Chung, Seo Hyun Kim, Sang Won Han, Joong Hyun Park, Jinkwon Kim, Yo Han Jung, Han-Jin Cho, Seong Hwan Ahn, Sung Ik Lee, Kwon-Duk Seo, Yoonkyung Chang, Tae-Jin Song, Hyo Suk Nam

**Affiliations:** 1Department of Neurology, Seoul Hospital, College of Medicine, Ewha Woman’s University, Seoul 07804, Korea; khhhj7@naver.com (H.J.K.); pierceu@hanmail.net (M.-S.P.); 2Department of Neurology, Yongin Severance Hospital, Yonsei University College of Medicine, Yongin 16995, Korea; quarksea@gmail.com; 3Department of Neurology, Yonsei University College of Medicine, Seoul 03722, Korea; neuro05@yuhs.ac (Y.D.K.); jnheo@yuhs.ac (J.H.); antithrombus@gmail.com (J.K.); 4Department of Neurology, Keimyung University School of Medicine, Daegu 42601, Korea; hjpark209042@gmail.com (H.P.); sungil.sohn@gmail.com (S.-I.S.); neurohong79@gmail.com (J.-H.H.); 5Department of Radiology, Yonsei University College of Medicine, Seoul 03722, Korea; bmoon21@yuhs.ac (B.M.K.); djkimmd@yuhs.ac (D.J.K.); 6Department of Neurology, Samsung Medical Center, Sungkyunkwan University School of Medicine, Seoul 06351, Korea; ohyoung.bang@samsung.com (O.Y.B.); mcastenosis@gmail.com (W.-K.S.); neurocjw@gmail.com (J.-W.C.); 7Department of Preventive Medicine, Yonsei University College of Medicine, Seoul 03722, Korea; hckim@yuhs.ac; 8College of Pharmacy, Yonsei Institute for Pharmaceutical Research, Yonsei University, Incheon 21983, Korea; eunahan@yonsei.ac.kr; 9Department of Neurology, Seoul Medical Center, Seoul 02053, Korea; gumicjg@naver.com; 10Department of Neurology, Gangnam Severance Hospital, Yonsei University College of Medicine, Seoul 06273, Korea; kylee@yuhs.ac (K.-Y.L.); yhjung@yuhs.ac (Y.H.J.); 11Biostatistics Collaboration Unit, Department of Research Affairs, Yonsei University College of Medicine, Seoul 03722, Korea; HSLEE1@yuhs.ac; 12Department of Neurology, Gachon University Gil Medical Center, Incheon 21565, Korea; dr.donghoon.shin@gmail.com; 13Department of Neurology, Kyung Hee University Hospital at Gangdong, Kyung Hee University School of Medicine, Seoul 05278, Korea; hyechoi@gmail.com; 14Department of Neurology, National Medical Center, Seoul 04564, Korea; jjongyl@gmail.com; 15Department of Neurology, Kangbuk Samsung Hospital, Sungkyunkwan University School of Medicine, Seoul 03181, Korea; janghyun.baek@gmail.com; 16Department of Neurology, National Health Insurance Service Ilsan Hospital, Goyang 10444, Korea; myoungsim@naver.com (G.S.K.); seobin7@naver.com (K.-D.S.); 17Department of Neurology, Yonsei University Wonju College of Medicine, Wonju 26426, Korea; s-hkim@yonsei.ac.kr; 18Department of Neurology, Sanggye Paik Hospital, Inje University College of Medicine, Seoul 01757, Korea; sah1puyo@gmail.com (S.W.H.); truelove1@hanmail.net (J.H.P.); 19Department of Neurology, Pusan National University School of Medicine, Busan 49241, Korea; chohj75@pusan.ac.kr; 20Department of Neurology, Chosun University School of Medicine, Gwangju 61453, Korea; shahn@Chosun.ac.kr; 21Department of Neurology, Sanbon Hospital, Wonkwang University School of Medicine, Gunpo 15865, Korea; neurologist@hanmail.net; 22Department of Neurology, Mokdong Hospital, College of Medicine, Ewha Woman’s University, Seoul 07985, Korea; tin1207@nate.com

**Keywords:** ischemic stroke, stroke risk score, recanalization, thrombectomy

## Abstract

Background: The CHADS_2_, CHA_2_DS_2_-VASc, ATRIA, and Essen scores have been developed for predicting vascular outcomes in stroke patients. We investigated the association between these stroke risk scores and unsuccessful recanalization after endovascular thrombectomy (EVT). Methods: From the nationwide multicenter registry (Selection Criteria in Endovascular Thrombectomy and Thrombolytic therapy (SECRET)) (Clinicaltrials.gov NCT02964052), we consecutively included 501 patients who underwent EVT. We identified pre-admission stroke risk scores in each included patient. Results: Among 501 patients who underwent EVT, 410 (81.8%) patients achieved successful recanalization (mTICI ≥ 2b). Adjusting for body mass index and *p* < 0.1 in univariable analysis revealed the association between all stroke risk scores and unsuccessful recanalization (CHADS_2_ score: odds ratio (OR) 1.551, 95% confidence interval (CI) 1.198–2.009, *p* = 0.001; CHA_2_DS_2_VASc score: OR 1.269, 95% CI 1.080–1.492, *p* = 0.004; ATRIA score: OR 1.089, 95% CI 1.011–1.174, *p* = 0.024; and Essen score: OR 1.469, 95% CI 1.167–1.849, *p* = 0.001). The CHADS_2_ score had the highest AUC value and differed significantly only from the Essen score (AUC of CHADS_2_ score; 0.618, 95% CI 0.554–0.681). Conclusion: All stroke risk scores were associated with unsuccessful recanalization after EVT. Our study suggests that these stroke risk scores could be used to predict recanalization in stroke patients undergoing EVT.

## 1. Introduction

Endovascular thrombectomy (EVT) plays a pivotal role in improving the prognosis by recanalizing occluded blood vessels in stroke patients [1]. With the recent success of trials on EVT, the number of patients receiving EVT continues to increase [1,2,3,4,5]. Moreover, the time window for EVT has also expanded [4,5]. Nevertheless, a significant number of patients who underwent EVT did not achieve successful recanalization [6]. As unsuccessful recanalization predictably leads to poor patient prognosis, it is important to identify the factors associated with unsuccessful recanalization. Factors associated with such unsuccessful recanalization include greater age, stroke severity, occlusion due to atherosclerosis, and thrombus burden [7,8,9]. Nonetheless, further research is still needed to identify the factors involved in unsuccessful recanalization [7,8,9]. 

Several stroke risk scores have been developed for predicting the clinical outcome or the occurrence of stroke. The CHADS_2_ [10], CHA_2_DS_2_-VASc [11], and ATRIA scores [12] are mainly used to predict thromboembolic risk and vascular outcome in atrial fibrillation (AF) patients. The Essen stroke risk score predicts vascular events in patients without AF [13]. As these stroke risk scores are mainly composed of risk factors and easily identifiable laboratory findings, they have the advantage of being able to easily predict the occurrence of stroke or prognosis. 

We hypothesized that stroke risk scores would be associated with unsuccessful recanalization in patients undergoing EVT. Hence, the purpose of this study was to investigate the association between increased CHADS_2_, CHA_2_DS_2_-VASc, ATRIA, and Essen scores and the results of recanalization after EVT.

## 2. Methods

### 2.1. Study Population

Our study included patients from the Selection Criteria in Endovascular Thrombectomy and Thrombolytic therapy (SECRET) registry (Clinicaltrials.gov NCT02964052). The selection criteria and the definition of included variables in this registry have been published [14]. In brief, the SECRET registry is a nationwide, multicenter registry that included patients undergoing reperfusion therapy such as EVT [14]. The SECRET registry did not establish strict inclusion or exclusion criteria for reperfusion therapy and recommended treatment according to the updated guideline at the time of treatment. Furthermore, the doctor of each institution determined whether to administer reperfusion therapy, and all patients who underwent reperfusion therapy were consecutively registered in the SECRET registry. All registered clinical and imaging information was reinvestigated and rechecked by the core laboratory after the anonymization process. The demographic data, risk factors for cardiovascular disease, medication history of prior index stroke, blood and urine laboratory examination results, time parameters for reperfusion therapy, neurologic status including severity, and image findings related to reperfusion therapy were investigated. 

Between January 2012 and December 2017, we retrospectively enrolled patients who received reperfusion thrombolysis and were consecutively registered in 15 hospitals. In addition, between November 2016 and December 2017, we prospectively enrolled patients who received reperfusion thrombolysis from 13 hospitals. A total of 1231 patients who underwent reperfusion thrombolysis were included, of which 507 patients underwent EVT. Finally, 501 patients who underwent EVT were included, excluding 6 patients, for whom information about the modified thrombolysis in cerebral infarction (mTICI) grade was not acquired (Figure 1). Written informed consent was obtained from the prospectively included patients or their next caregivers. Our Institutional Review Board approved our study (Yonsei University College of Medicine, 4-2015-1196).

Stroke severity was defined using the National Institutes of Health Stroke Scale (NIHSS) score, and the neurologic change after 24 h of EVT was defined as the difference between the initial NIHSS score and the NIHSS score at 24 h (Initial NIHSS score—NIHSS score at 24 h = change in NIHSS score after 24 h). Therefore, if this value was positive, it means neurological improvement, 0 means no improvement, and negative means neurological worsened. Time parameters of EVT were acquired from onset-to-start of EVT (onset to puncture time) and administration of intravenous (IV) thrombolysis (tissue plasminogen activator, tPA) to start of EVT (needle to puncture time) [14]. In case of unclear symptom onset time, the last normal time (LNT) when the patient was asymptomatic was considered as the time of onset. Computed tomography (CT), CT angiography, magnetic resonance imaging (MRI), MR angiography, and digital subtraction angiography (DSA) images were acquired during the admission period. 

Data related to reperfusion therapy, for example, the administration of IV thrombolysis, the total trial number of stent-retriever passes and the types of devices were investigated. Intra-arterial (IA) thrombolysis without IV tPA is defined as first-line therapy with EVT, who are contraindicated for IV tPA. Combined IV/IA thrombolysis is defined as IV tPA administration prior to EVT who could be treated with IV tPA within 4.5 h after symptom onset. The status of reperfusion therapy was investigated in the patients who underwent EVT using the final angiographic findings, including the DSA, and graded based on the mTICI grade. For the outcome parameter, a grade of mTICI 2b or 3 was defined as successful recanalization, and a grade of mTICI 0–2a was defined as unsuccessful recanalization. EVT was performed using a stent-retriever technique, a direct aspiration first pass technique (ADAPT), and the Solumbra technique. The first-line technique is based on the clinical situation of each center and each patient. If the first-line technique is unsuccessful, the second-line technique is used. Stent-retriever alone was defined as using only a stent-retriever as a first-line technique and not using ADAPT or the Solumbra technique as the second-line technique. Aspiration alone was defined as using ADAPT as a first-line technique and not using any other device as a second-line technique. The type of device used for each technique was based on operator preference (typically Solitaire FR device, Trevor stent device, and Penumbra). 

### 2.2. The Stroke Risk Scoring Systems

We identified pre-admission CHADS_2_, CHA_2_DS_2_-VASc, ATRIA, and Essen scores for each patient. The variables included in each scoring system are set according to the existing definition. The CHADS_2_ and CHA_2_DS_2_-VASc scores, congestive heart failure, hypertension, age, diabetes mellitus (DM), previous stroke history, vascular diseases, and sex were included as scoring variables [10,11]. The ATRIA score included age, sex, hypertension, DM, congestive heart failure, presence of proteinuria, and kidney dysfunction (estimated glomerular filtration rate <45 mL/min per 1.73 m^2^) as scoring parameters [12]. The Essen score included age, hypertension, DM, previous stroke history, myocardial infarction history, peripheral arterial occlusive disease, and other vascular diseases [13]. 

### 2.3. Statistical Analyses

Continuous variables and categorical variables were analyzed using an independent *t*-test or Mann–Whitney *U* test and the chi-square test or Fisher’s exact test, respectively. Uni- and multivariable logistic regression was performed to evaluate factors for unsuccessful recanalization. Body mass index (BMI) and onset to puncture time, which are important cofounders for unsuccessful recanalization, and *p* < 0.1 (excluding age and DM, which are common overlapping variables for all stroke risk scores) from the univariable analysis were entered in multivariable analysis. The results of uni- and multivariable analyses were expressed as odds ratios (ORs) and 95% confidence intervals (CIs). Because the risk of vascular outcome increased as the stroke risk scores increased, the main outcome was defined as unsuccessful recanalization in this study. Subgroup analyses were performed, including demographic data, classical vascular risk factors, and stroke risk scores, and were dichotomized by the median values and the optimal cut off values. The interaction between unsuccessful recanalization and each subgroup was investigated with a two-tailed test in the logistic regression analyses. For the sensitivity analysis, we further analyzed all stroke risk scores for patients with AF-related stroke only. 

For evaluating the predictability of CHADS_2_, CHA_2_DS_2_-VASc, ATRIA, and Essen scores, receiver operating characteristic (ROC) curve analysis and area under the curve (AUC) were investigated. The AUC was calculated and the optimal cutoff values of the stroke risk scores were defined at the level with the highest Youden index (sensitivity + specificity − 1). The AUC of each stroke risk score was compared to determine whether there was a difference in the predictability of unsuccessful recanalization among the stroke risk scores. We utilized the multivariable model as the benchmark to assess the role of stroke risk scores in enhancing the risk prediction for unsuccessful recanalization in EVT patients. We compared AUCs to assess model discrimination and calculated net reclassification improvement (NRI) and the integrated discrimination improvement (IDI). All statistical analyses were performed using SPSS (version 25.0, IBM Corp., Chicago, IL, USA) and open-source statistical package R version 3.6.3 (R Project for Statistical Computing, Vienna, Austria). All variables needed a *p* < 0.05 to be considered statistically significant.

## 3. Results

### 3.1. Study Population

A total of 501 patients were included in this study. Patient demographics and information on risk factors and variables are summarized in Table 1. Of the 501 patients receiving EVT, 234 patients (46.7%) were female, and the mean age was 76.2 ± 13.3 years. The median value of the NIHSS scores of all patients was 15 (10–19, interquartile range (IQR)). IV thrombolysis was administered to 202 patients (40.3%), and the mean value of the onset to needle time was 119.9 ± 97.3 min. In all patients who underwent EVT, the mean value of the onset to puncture time was 354.6 ± 440.0 min, the mean value of the needle to puncture time was 78.7 ± 50.8 min, stent-retriever alone was used in 371 patients (74.0%), aspiration alone in 25 patients (4.9%), and combined stent-retriever and aspiration in 90 patients (17.9%). Among the patients who underwent stent-retriever alone and combined stent-retriever/aspiration, information about the stent device was obtained from 440 patients: the Solitaire FR device was used in 377 (85.6%) patients, the Trevor stent device in 58 (13.1%) patients, and both stent devices in only 5 (1.1%) patients. The mean value of the number of stent-retriever passes was 2.1 ± 1.9. Among the patients who underwent aspiration alone and combined stent-retriever/aspiration, aspiration device information was obtained from 92 patients: the Penumbra aspiration system was used in 55 (59.7%) patients and an intermediate catheter device in 37 (40.2%) patients.

Among all included patients, 410 (81.8%) patients achieved successful recanalization (mTICI ≥ 2b). The onset to recanalization measured only for patients who successfully recanalized (mTICI 2b/3) was 429.5 ± 481.4 min. 

### 3.2. Association of Stroke Risk Scores with Recanalization Status

In the successful recanalization group, the proportion of patients with DM was lower (53.1% vs. 70.3%, *p* = 0.004), and there were more patients with coronary disease (31.2% vs. 17.5%, *p* = 0.013). Patients in the successful recanalization group had lower initial NIHSS scores (median 15 (IQR 10–19) vs. median 17 (IQR 12–20.5), *p* = 0.020) and the change in NIHSS scores after 24 h was greater (Initial NIHSS score—NIHSS score at 24 h, median 5 (IQR 0–10) vs. median 0 (IQR −2–3), *p* < 0.001) than those in the unsuccessful recanalization group. Combined IA/IV thrombolysis was significantly associated with successful recanalization (*p* = 0.030). In patients who administration of tPA prior to EVT, the time interval of the needle to puncture was significantly shorter in the successful recanalization group (111.8 ± 50.1 vs. 73.6 ± 49.1, *p* < 0.001). The stent-retriever alone was associated with successful recanalization (*p* = 0.035). However, aspiration alone (*p* = 0.035) was associated with unsuccessful recanalization. In patients who received stent-retrievers, the number of stent passes was significantly lower in the successful recanalization group (2.9 ± 2.8 vs. 2.0 ± 1.6, *p* = 0.002). In laboratory tests, both initial glucose level after admission (152.9 ± 54.3 mg/dL vs. 140.2 ± 49.7 mg/dL, *p* = 0.042) and fasting glucose level after admission (148.8 ± 52.9 mg/dL vs. 128.9 ± 46.9 mg/dL, *p* = 0.002) were lower in the successful recanalization group. All stroke risk scores were significantly lower in the successful recanalization group (CHADS_2_ score; median 2 (IQR 1–3) vs. 3 (IQR 2–3), *p* < 0.001) (CHA_2_DS_2_VASc score; median 3 (IQR 2–4] vs. 4 (IQR 3–5], *p* = 0.002) (ATRIA score; median 7 (IQR 3–9) vs. 9 (IQR 6–10), *p* = 0.002) (Essen score; median 3 (IQR 2–4) vs. 4 (IQR 3–4), *p* = 0.034) (Table 1).

In univariable logistic regression analysis, age, DM, coronary disease, initial NIHSS score, combined IA/IV thrombolysis, stent-retriever alone, aspiration alone, number of stent-retriever passes, and stroke risk scores were associated with unsuccessful recanalization, as shown in Appendix A. In multivariable logistic regression analysis, all stroke risk scores were predictive of unsuccessful recanalization along with BMI, onset to puncture time, coronary disease, initial NIHSS score, combined IA/IV thrombolysis, stent-retriever alone, aspiration alone, and the number of stent-retriever passes (CHADS_2_ score: OR 1.551, 95% CI 1.198–2.009, *p* = 0.001; CHA_2_DS_2_VASc score: OR 1.269, 95% CI 1.080–1.492, *p* = 0.004; ATRIA score: OR 1.089, 95% CI 1.011–1.174, *p* = 0.024; and Essen score: OR 1.469, 95% CI 1.167–1.849, *p* = 0.001) (Table 2). 

In subgroup analysis, the association of unsuccessful recanalization was stratified by age, sex, comorbidities, NIHSS score, treatment factor (LNT, combined IA/IV thrombolysis), and stroke risk scores. A subgroup of patients with DM (*p* for interaction = 0.003), IA alone (*p* for interaction = 0.023), CHA_2_DS_2_VASc score ≥ 4 (*p* for interaction = 0.022), ATRIA score ≥ 8 (*p* for interaction = 0.017), CHADS_2_ score ≥ 3 (*p* for interaction < 0.001), CHA_2_DS_2_VASc score ≥ 5 (*p* for interaction < 0.001), ATRIA score ≥ 9 (*p* for interaction = 0.001), and Essen score ≥ 4 (*p* for interaction = 0.009) were significantly associated with unsuccessful recanalization (Figure 2).

In the comparison with AF-related stroke, there were significantly fewer patients with DM (72.1% vs. 53.6%, *p* = 0.042) and more patients with stent-retriever alone (51.2% vs. 80.1%, *p* < 0.001) in the successful recanalization group. Moreover, patients in the successful recanalization group had lower initial NIHSS scores (median 15 (IQR 10–19) vs. median 17 (IQR 12–20.5), *p* = 0.020) and the change in NIHSS scores after 24 h was greater (Initial NIHSS score—NIHSS score at 24 h, median 5 (IQR 1–10) vs. median 0 (IQR −1–2), *p* < 0.001) than those in the unsuccessful recanalization group. An increase in the number of stent-retriever passes was associated with unsuccessful recanalization (3.3 ± 3.0 vs. 2.0 ± 1.5, *p* = 0.007). In laboratory tests, fasting glucose after admission (146.24 ± 50.3 mg/dL vs. 128.4 ± 51.9 mg/dL, *p* = 0.044) were lower in the successful recanalization group. The CHADS_2_, CHA_2_DS_2_VASc, ATRIA, and Essen scores were significantly lower in the successful recanalization group (CHADS_2_ score; median 2 (IQR 2–3) vs. 3 (IQR 2–3), *p* < 0.001) (CHA_2_DS_2_VASc score; median 4 (IQR 3–4] vs. 5 (IQR 4–5.5], *p* = 0.003) (ATRIA score; median 8 (IQR 6–9) vs. 9 (IQR 7.5–10), *p* = 0.033) (Essen score; median 4 (IQR 3–4) vs. 3 (IQR 3–4), *p* = 0.043). The above results are summarized in Appendix A. In multivariable logistic regression analysis, the CHADS_2_, CHA_2_DS_2_VASc, and Essen scores were associated with unsuccessful recanalization along with BMI, coronary disease, the initial NIHSS score, combined IA/IV thrombolysis, stent-retriever alone, aspiration alone, and number of stent-retriever passes (CHADS_2_ score: OR 1.787, 95% CI 1.173–2.725, *p* = 0.007; CHA_2_DS_2_VASc score: OR 1.354, 95% CI 1.049–1.747, *p* = 0.020; and Essen score: OR 1.635, 95% CI 1.093–2.448, *p* = 0.017) (Appendix A).

### 3.3. Comparison of Stroke Risk Scores for Unsuccessful Recanalization

Figure 3 shows the ROC curves of all stroke risk scores for unsuccessful recanalization. The AUC, optimal cutoff value, sensitivity, specificity, positive predictive value, and negative predictive value of each stroke risk score are presented in Table 3. 

Among stroke risk scores, the CHADS_2_ score had the highest AUC value. However, in pairwise comparisons of the AUC, only the CHADS_2_ and Essen scores were significantly different (AUC of CHADS_2_ score; 0.618, 95% CI 0.554–0.681 vs. AUC of Essen score; 0.569, 95% CI 0.506–0.632, *p* = 0.002) (Appendix A). Similarly, even when ROC curve analysis was performed on AF-related stroke only, the CHADS_2_ score had the highest AUC value, and only the CHADS_2_ and Essen scores were significantly different (AUC of CHADS_2_ score; 0.666, 95% CI 0.570–0.761 vs. AUC of Essen score; 0.595, 95% CI 0.502–0.687, *p* = 0.006) (Appendix A). 

The continuous-based NRI was significantly improved after the addition of each stroke risk score (CHADS_2_ score: *p* = 0.010, CHA_2_DS_2_VASc score: *p* = 0.023, ATRIA score: *p* = 0.035, Essen score: *p* = 0.005). The IDI also showed improved risk classification after the addition of the CHADS_2_ score (*p* = 0.014) or the ATRIA score (*p* = 0.015). Overall, the best model for prediction of unsuccessful recanalization after EVT was the CHADS_2_ score, with the addition of the multivariable model (Appendix A).

## 4. Discussion

The key finding of this study was that the pre-admission CHADS_2_, CHA_2_DS_2_VASc, ATRIA, and Essen scores were associated with unsuccessful recanalization after EVT. The probability of unsuccessful recanalization increased as the stroke risk scores increased. The CHADS2 score had the highest AUC among all stroke risk scores, although the CHADS2 score differed significantly only from the Essen score. 

Previous studies have proven the relationship between stroke risk scores and the clinical outcome of stroke. CHADS_2_, CHA_2_DS_2_-VASc, and ATRIA scores are simple to obtain and are useful tools for estimating the thromboembolic risk and clinical outcomes in patients with AF [15,16,17,18]. The Essen score is a simple clinical score that was derived to predict the 1-year risk of recurrent ischemic stroke after ischemic stroke based on the presence of prior vascular comorbidities [13,19]. Unlike the purpose for which the stroke risk scores were developed, the stroke risk scores have been used as a predictor of various outcomes in various patient groups [15,20,21]. Our results are meaningful in that they provide additional information that CHADS_2_, CHA_2_DS_2_-VASc, ATRIA, and Essen scores were correlated with unsuccessful recanalization in patients undergoing EVT, as well as thromboembolic risk and clinical outcome. All stroke risk scores were associated with unsuccessful recanalization even in AF-related stroke patients. 

Hypertension and DM have a common weighting factor for all stroke risk scores, and DM, in particular, is known to affect recanalization after IV thrombolysis [22,23]. Although DM and fasting hyperglycemia are also known to affect clinical outcomes after EVT [24,25], there is still insufficient evidence that DM and initial and fasting hyperglycemia influence recanalization after EVT [26,27]. Our results showed that DM, initial, and fasting glucose levels were associated with unsuccessful recanalization. Factors other than hypertension and DM were weighted differently for each stroke risk score. Compared with the CHADS_2_ score, the CHA_2_DS_2_-VASc score has a higher weighting for age and includes the components of sex and vascular diseases; the ATRIA score has a higher weighting for age and includes the components of sex and chronic renal disease, while the Essen score also has a higher weighting for age and sex, similarly to the other scores, along with weighting for vascular disease and current smoking. Each additional component of these scores has been reported as a predictor of stroke severity or outcome [28]. The ATRIA and CHA_2_DS_2_-VASc scores were reported to outperform the CHADS_2_ score in predicting stroke outcome in patients with AF [17,28]. However, unlike previous studies that investigated the relationship between the stroke risk scores and stroke outcome, we found that the CHADS_2_ score shows better performance in predicting recanalization than the CHA_2_DS_2_-VASc and ATRIA scores in EVT patients. This may be because sex, chronic kidney disease, and vascular disease weighted by the CHA_2_DS_2_-VASc and ATRIA scores did not differ with recanalization, and there were more patients with coronary disease in the successful recanalization group in our dataset. As the Essen score is a risk-scoring tool for the prediction of recurrent stroke and combined cerebrovascular events in patients with non-AF, there have been a few studies comparing the performance of the Essen score with that of the CHADS_2_ score [13]. A recent observational study of the prediction for vascular outcome in stroke patients with AF found no significant difference in the performance between the CHADS_2_ and Essen scores [21]. In contrast, the CHADS_2_ score showed significantly better performance than the Essen score in our study. Even when only AF-related stroke patients were analyzed, the Essen score could significantly predict unsuccessful recanalization, although its performance was worse than that of the CHADS_2_ score. Presumably, as in the case of CHA_2_DS_2_-VASc and ATRIA scores, this could be attributed to the observation that peripheral artery disease and current smoking weighted by the Essen score did not differ according to recanalization, and there were more patients with coronary disease in the recanalization group in our dataset. Therefore, unlike previous studies on patients with AF stroke, the ATRIA and CHA_2_DS_2_-VASc scores likely did not outperform and the Essen score likely underperformed compared to the CHADS_2_ score. The significance of this result suggests that most of the factors related to unsuccessful recanalization in EVT patients overlap with most factors related to the occurrence of stroke in AF patients. Therefore, these different factors should be taken into account when creating a new scoring system that predicts recanalization after EVT. An existing pre-admission stroke risk score or suitable new scoring system can be used in addition to the current image-based patient’s selection system, which can contribute to lower recanalization failure rates by appropriately selecting patients.

### Limitations

First, although some of the patients included in our study were prospectively included and the registry itself consecutively included stroke patients who received reperfusion therapy, we performed a retrospective evaluation. Therefore, there may be selection bias, l and the possibility of a causal relationship cannot be concluded. Second, this registry is a nationwide observational registry that reflects real-world evidence; however, there may be a selection bias because it is not a randomized controlled study. To reduce the selection bias, we consecutively included patients eligible for EVT according to the valid guidelines [1,29,30]. Third, because our registry enrolled only the Korean population, it is difficult to generalize our findings to all races.

## 5. Conclusions

The pre-admission CHADS_2_, CHA_2_DS_2_VASc, ATRIA, and Essen scores were associated with unsuccessful recanalization after EVT. Therefore, these results suggest that stroke risk scores, especially the CHADS_2_ score, could predict recanalization in stroke patients undergoing EVT. 

## Figures and Tables

**Figure 1 jcm-11-00274-f001:**
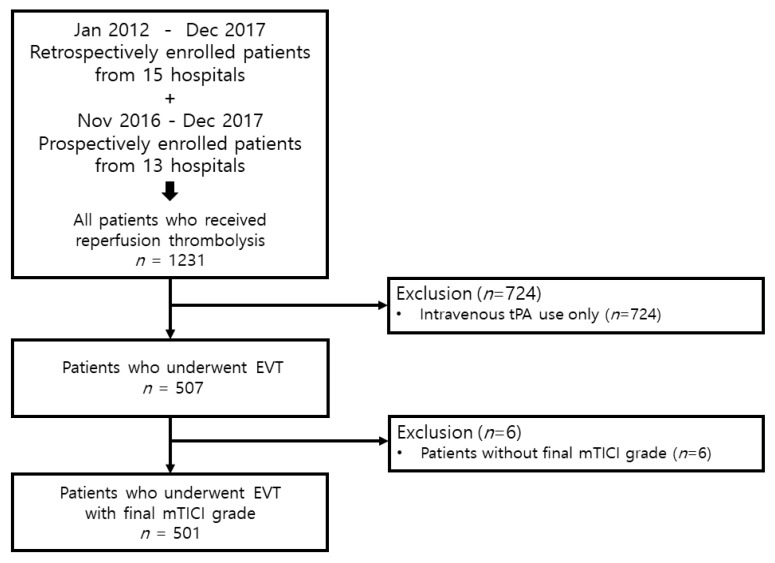
Patient selection strategy used in the study. tPA, tissue plasminogen activator; EVT, endovascular thrombectomy; mTICI, modified thrombolysis in cerebral infarction.

**Figure 2 jcm-11-00274-f002:**
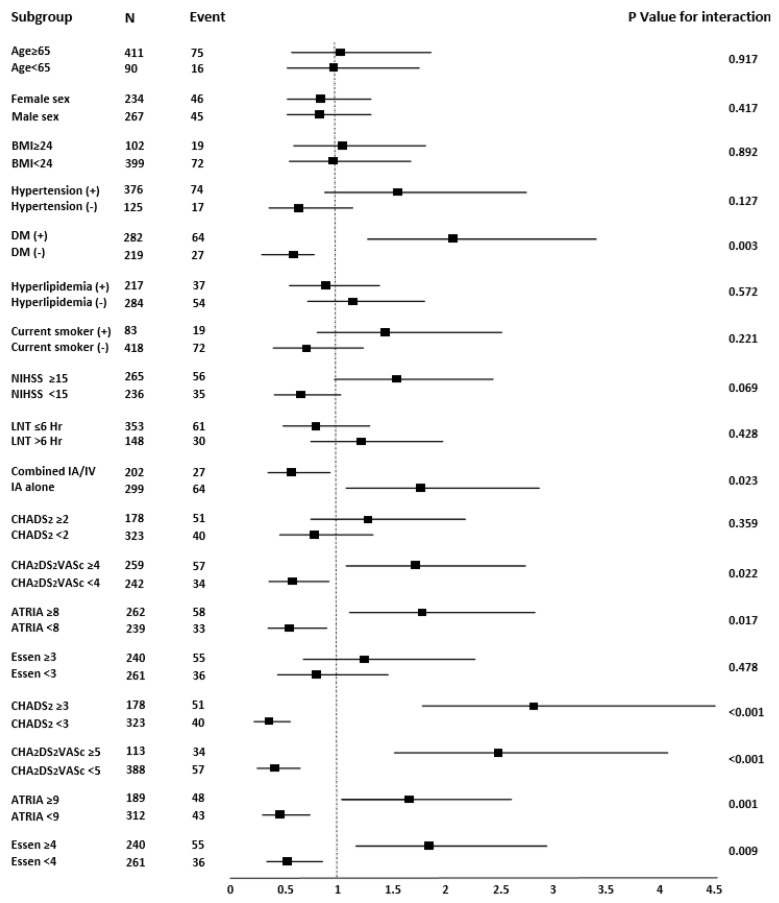
Forest plots of unadjusted odds ratios for unsuccessful recanalization (mTICI ≤ 2a) in patients with endovascular thrombectomy. BMI, body mass index; NIHSS, National Institutes of Health Stroke Scale; LNT, last normal time; IA, intra-arterial; IV, intra-venous; ORs, odds ratios; mTICI, modified thrombolysis in cerebral infarction.

**Figure 3 jcm-11-00274-f003:**
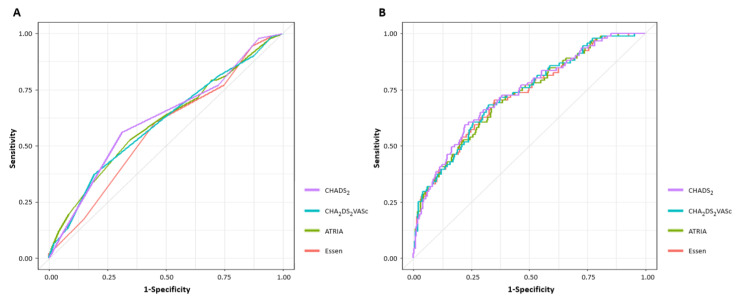
Receiver operating characteristic curve analyses of unsuccessful recanalization based on stroke risk scores. (**A**) Univariable ROC analysis (**B**) Multivariable ROC analysis. ROC, receiver operating characteristic.

**Table 1 jcm-11-00274-t001:** Clinical and imaging characteristics according to the degree of recanalization.

	Total (*n* = 501)	Unsuccessful Recanalization mTICI ≤ 2a (*n* = 91)	Succeesful Recanalization mTICI 2b/3 (*n* = 410)	*p*-Value
Age, years, mean (SD)	76.2 ± 13.3	78.7 ± 14.1	75.6 ± 13.0	0.059
Female, (%)	234 (46.7%)	46 (50.6%)	188 (45.9%)	0.486
BMI (kg/m^2^), mean (SD)	20.6 ± 4.1	20.3 ± 4.8	20.6 ± 4.0	0.519
**Vascular risk factors**				
Hypertension, (%)	376 (75.1%)	74 (81.3%)	302 (73.7%)	0.163
Diabetes mellitus, (%)	282 (56.3%)	64 (70.3%)	218 (53.2%)	0.004
Hypercholesterolemia, (%)	217 (43.3%)	37 (40.7%)	180 (43.9%)	0.654
Current smoking, (%)	83 (16.6%)	19 (20.9%)	64 (15.6%)	0.286
eGFR < 60 mL/min, (%)	243 (48.5%)	51 (56.0%)	192 (46.8%)	0.140
**Comorbidities**				
Atrial fibrillation (%)	265 (52.9%)	51 (56.0%)	214 (52.2%)	0.583
Heart failure, (%)	43 (8.6%)	11 (12.1%)	32 (7.8%)	0.266
Coronary disease, (%)	144 (28.7%)	16 (17.6%)	128 (31.2%)	0.013
Peripheral artery disease, (%)	17 (3.4%)	6 (6.6%)	11 (2.7%)	0.123
Previous infarction, (%)	118 (23.6%)	23 (25.3%)	95 (23.2%)	0.771
Previous hemorrhage, (%)	27 (5.4%)	7 (7.7%)	20 (4.9%)	0.413
**Medication before admission**				
Prior antiplatelet therapy, (%)	156 (31.1%)	29 (31.9%)	127 (31.0%)	0.967
Prior anticoagulant therapy, (%)	88 (17.6%)	12 (13.2%)	76 (18.5%)	0.053
Prior statin therapy, (%)	152 (30.3%)	21 (23.1%)	131 (32.0%)	0.124
Initial NIHSS score, median (IQR)	15 (10–19)	17 (12–20.5)	15 (10–19)	0.020
Change in NIHSS score after 24 h, median (IQR)	4 (0–9)	0 (−2–3)	5 (0–10)	<0.001
**Treatment**				
IA thrombolysis without IV tPA, (%)	299 (59.7%)	64 (70.3%)	235 (57.3%)	0.030
Combined IV/IA thrombolysis *, (%)	202 (40.3%)	27 (29.7%)	175 (42.7%)	0.030
Stent-retriever alone, (%)	371 (74.1%)	52 (57.1%)	319 (77.8%)	<0.001
Aspiration alone, (%)	25 (5.0%)	9 (9.9%)	16 (3.9%)	0.035
Combined stent-retriever/aspiration **, (%)	90 (18.0%)	15 (16.5%)	75 (18.3%)	0.521
Number of stent-retriever passes, mean (SD)	2.1 ± 1.9	2.9 ± 2.8	2.0 ± 1.6	0.002
Onset to puncture, min, mean (SD)	354.6 ± 440.0	370.1 ± 293.5	351.1 ± 466.6	0.621
LNT-to-puncture time (within 6 h)	350 (69.9%)	60 (65.9%)	290 (70.7%)	0.438
**Arterial occlusion site**				
Any ICA, (%)	94 (18.8%)	16 (17.6%)	78 (19.0%)	0.865
MCA, (%)	127 (25.4%)	18 (19.8%)	109 (26.6%)	0.224
ACA, (%)	7 (1.4%)	1 (1.1%)	6 (1.5%)	>0.99
PCA, (%)	8 (1.6%)	1 (1.1%)	7 (1.7%)	0.259
V-B, (%)	40 (8.0%)	10 (11.0%)	30 (7.3%)	0.340
Tandem lesion	24 (4.8%)	5 (5.5%)	19 (4.6%)	0.939
**Stroke etiology**				0.380
Cardioembolic	270 (53.9%)	49 (53.9%)	221 (53.9%)	
Large artery atherosclerosis	83 (16.6%)	19 (20.9%)	64 (15.6%)	
Undetermined or others	148 (29.5%)	23 (25.3%)	125 (30.5%)	
**Laboratory results**				
Initial glucose ^†^, mg/dL	145.3 ± 51.4	152.9 ± 54.3	140.2 ± 49.7	0.042
Fasting glucose ^‡^, mg/dL	135.5 ± 51.7	148.8 ± 52.9	128.9 ± 46.9	0.002
**Pre-admission stroke risk score,** **score, median (IQR)**				
CHADS_2_ score	2.1 ± 1.0	3 (2–3)	2 (1–3)	<0.001
CHA_2_DS_2_VASc score	3.4 ± 1.6	4 (3–5)	3 (2–4)	0.002
ATRIA score	6.7 ± 3.6	9 (6–10)	7 (3–9)	0.002
Essen score	3.3 ± 1.4	4 (3–4)	3 (2–4)	0.034

mTICI, modified thrombolysis in cerebral infarction; SD, standard deviation; BMI, body mass index; eGFR, estimated glomerular filtration rate; National Institutes of Health Stroke Scale, NIHSS; IQR, interquartile range; tPA, tissue plasminogen activator; IA, int; IV, intravenous; LNT, last normal time; ICA, internal carotid artery; MCA, middle cerebral artery; ACA, anterior cerebral artery; PCA, posterior cerebral artery; V-B, vertebro-basilar. * administration of intravenous tissue plasminogen activator prior to endovascular thrombectomy; ** cases in which stent-retriever and aspiration were performed simultaneously or sequentially. ^†^ The glucose level test was performed at the time of the first admission to the emergency room. ^‡^ The glucose level test was performed after 8 h of fasting after admission.

**Table 2 jcm-11-00274-t002:** Multivariable analysis for stroke risk score associated with the unsuccessful recanalization among 501 patients with endovascular thrombectomy.

	CHADS_2_		CHA_2_DS_2_VASc		ATRIA		Essen	
Variables	OR (95% CI)	*p*-Value	OR (95% CI)	*p*-Value	OR (95% CI)	*p*-Value	OR (95% CI)	*p*-Value
BMI, per-1-kg/m^2^ increase	0.994 (0.936–1.055)	0.836	1.007 (0.947–1.071)	0.986	1.003 (0.943–1.066)	0.937	0.986 (0.928–1.047)	0.612
Coronary disease	0.372 (0.200–0.691)	0.002	0.383 (0.206–0.710)	0.002	0.380 (0.205–0.704)	0.002	0.251 (0.126–0.499)	<0.001
Initial NIHSS score, per 1-score increase	1.015 (0.977–1.055)	0.448	1.017 (0.979–1.057)	0.424	1.016 (0.978–1.056)	0.412	1.017 (0.978–1.056)	0.376
**IV thrombolysis**								
IA thrombolysis without IV tPA	Reference		Reference		Reference		Reference	
Combined IA/IV thrombolysis *	0.647 (0.382–1.094)	0.104	0.654 (0.388–1.105)	0.113	0.674 (0.399–1.140)	0.142	0.625 (0.370–1.055)	0.079
**EVT parameters**								
Stent-retriever alone	0.528 (0.302–0.924)	0.025	0.547 (0.313–0.957)	0.035	0.512 (0.294–0.892)	0.018	0.536 (0.307–0.939)	0.029
Aspiration alone	2.966 (1.048–8.394)	0.041	3.344 (1.183–9.453)	0.023	3.142 (1.122–8.880)	0.029	2.887 (1.024–8.140)	0.045
Number of stent-retriever passes, per-1-passes increase	1.267 (1.124–1.428)	<0.001	1.274 (1.131–1.436)	<0.001	1.275 (1.132–1.436)	<0.001	1.267 (1.124–1.427)	<0.001
Onset to puncture, per 1-min increase	1.000 (0.999–1.001)	0.992	1.000 (1.000–1.001)	0.992	1.000 (1.000–1.001)	0.935	1.000 (1.000–1.001)	0.980
**Risk scoring score**								
Per-1-point increase	1.551 (1.198–2.009)	0.001	1.269 (1.080–1.492)	0.004	1.105 (1.027–1.188)	0.007	1.469 (1.167–1.849)	0.001

OR, odds ratio; CI, confidence interval; BMI, body mass index; National Institutes of Health Stroke Scale, NIHSS; IV, intravenous; IA, intra-arterial; tPA, tissue plasminogen activator; EVT, endovascular thrombectomy; * administration of intravenous tissue plasminogen activator prior to endovascular thrombectomy.

**Table 3 jcm-11-00274-t003:** Receiver operating characteristic (ROC) curve analysis of risk scores for the probability of an unsuccessful recanalization.

	AUC	Optimal Cutoff	Diagnostic Sensitivity	Diagnostic Specificity	PPV	NPV
CHADS_2_ score	0.618	2.5	0.560	0.690	0.287	0.876
CHA_2_DS_2_VASc score	0.602	4.5	0.374	0.807	0.301	0.853
ATRIA score	0.605	8.5	0.528	0.656	0.254	0.862
Essen score	0.569	3.5	0.549	0.604	0.862	0.229

AUC, area under the curve; PPV, positive predictive value; NPV, negative predictive value.

## Data Availability

The data presented in this study are available on request from the corresponding authors. The data are not publicly available due to privacy.

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
