# Peer review of "Association between CHADS2, CHA2DS2-VASc, ATRIA, and Essen Stroke Risk Scores and Unsuccessful Recanalization after Endovascular Thrombectomy in Acute Ischemic Stroke Patients"

_jcm, 2022, doi:10.3390/jcm11010274_

Round 1
Reviewer 1 Report
The authors responded satisfactorily to the comments from the previous review.
Author Response
Response to Reviewers’ comments
December 26, 2021
Manuscript ID: jcm-1427981
Title: Association between CHADS2, CHA2DS2-VASc, ATRIA, and Essen stroke risk scores and unsuccessful recanalization after endovascular thrombectomy in acute ischemic stroke patients
Dear reviewers & editor
First of all, we would like to thank the reviewers and editorial board members for their time and valuable comments on our manuscript. We have addressed our opinions on each comment from two reviewers in this response letter, and made several changes and corrections to our original manuscript.
We tried to revise our manuscript according to reviewer’s suggestions as much as possible, and second revised parts are written in blue color texts in the manuscript. We hope that the revisions in the manuscript and our accompanying responses will be sufficient to make our manuscript suitable for publication in Journal of Clinical Medicine.
We shall look forward to hearing from you at your earliest convenience.
Sincerely yours,
Tae-Jin Song, MD, PhD.
Department of Neurology, Seoul Hospital, Ewha Womans University College of Medicine, 260, Gonghang-daero, Gangseo-gu, Seoul, 07804, Republic of Korea
Tel: +82-2-2650-2677, Fax: +82-2-2650-5958; E-mail: knstar@ewha.ac.kr
Hyo Suk Nam, MD, PhD
Department of Neurology, Yonsei University College of Medicine
50-1 Yonsei-ro, Seodaemoon-gu, Seoul, 03722, Korea
Tel: 82-2-2228-1617, Fax: 82-2-393-0705; E-mail: hsnam@yuhs.ac
Reviewer 2 Report
Dear Authors,
I still have doubts about a couple of elements of this article. In Tab 1 the description "IA thrombolysis without tPA" was changed, I would like to ask for clarification what the authors meant? How to perform i.a. thrombolysis without tPA? Can I also ask for a detailed description of the "combined IV/IA thrombolysis" procedure (doesn't have to be in the text, can be in the explanations to the reviewer)? Also, there is an error in the description of this Table in the new text (IA;tPA)
In the paragraph "Association of stroke risk scores...", in the newly added text, I got lost when considering whether NIHSS decreased with successful recanalization. Because if the median decrease in NIHSS is 0, I guess it's worse than if the median decrease in NIHSS is 5? Am I right? I also get the impression that the same text is duplicated and can be found again in the text below Figure 2.
Please explain why diabetes was removed from Table 2?
Best Regards
Author Response
Response to Reviewers’ comments
December 26, 2021
Manuscript ID: jcm-1427981
Title: Association between CHADS2, CHA2DS2-VASc, ATRIA, and Essen stroke risk scores and unsuccessful recanalization after endovascular thrombectomy in acute ischemic stroke patients
Dear reviewers & editor
First of all, we would like to thank the reviewers and editorial board members for their time and valuable comments on our manuscript. We have addressed our opinions on each comment from two reviewers in this response letter, and made several changes and corrections to our original manuscript.
We tried to revise our manuscript according to reviewer’s suggestions as much as possible, and second revised parts are written in blue color texts in the manuscript. We hope that the revisions in the manuscript and our accompanying responses will be sufficient to make our manuscript suitable for publication in Journal of Clinical Medicine.
We shall look forward to hearing from you at your earliest convenience.
Sincerely yours,
Tae-Jin Song, MD, PhD.
Department of Neurology, Seoul Hospital, Ewha Womans University College of Medicine, 260, Gonghang-daero, Gangseo-gu, Seoul, 07804, Republic of Korea
Tel: +82-2-2650-2677, Fax: +82-2-2650-5958; E-mail: knstar@ewha.ac.kr
Hyo Suk Nam, MD, PhD
Department of Neurology, Yonsei University College of Medicine
50-1 Yonsei-ro, Seodaemoon-gu, Seoul, 03722, Korea
Tel: 82-2-2228-1617, Fax: 82-2-393-0705; E-mail: hsnam@yuhs.ac
Comment 1. I still have doubts about a couple of elements of this article. In Tab 1 the description "IA thrombolysis without tPA" was changed, I would like to ask for clarification what the authors meant? How to perform i.a. thrombolysis without tPA? Can I also ask for a detailed description of the "combined IV/IA thrombolysis" procedure (doesn't have to be in the text, can be in the explanations to the reviewer)? Also, there is an error in the description of this Table in the new text (IA;tPA)
Answer 1. We apologize for confusion after the revision of our manuscript. Combined IV/IA thrombolysis is a case of administering IV tPA prior to endovascular thrombectomy who could be treated with IV tPA within 4.5 hours after symptom onset. Patients eligible for IV tPA were treated according to the AHA-ASA and Korean clinical practice guidelines. IA thrombolysis without tPA is a case of endovascular thrombectomy as a first-line therapy, who are contraindicated for IV tPA. To avoid confusion, we revised our manuscript.
From
“Data related to reperfusion therapy, for example, the total trial number of stent-retriever passes and the types of devices were investigated.”
To
“Data related to reperfusion therapy, for example, the administration of IV thrombolysis, the total trial number of stent-retriever passes and the types of devices were investigated.”
At page 4 # paragraph 4 #, in Methods section
From
“IA thrombolysis without tPA”
To
“IA thrombolysis without IV tPA”
From
“administration of tissue plasminogen activator prior to endovascular thrombectomy.”
To
“administration of intravenous tissue plasminogen activator prior to endovascular thrombectomy.”
#Table 1, Table 2, Table S1, Table S2, and Table S3 in Results section
And added following sentence.
“Intra-arterial (IA) thrombolysis without IV tPA is defined as a first-line therapy with EVT, who are contraindicated for IV tPA. Combined IV/IA thrombolysis is defined as IV tPA administration prior to EVT who could be treated with IV tPA within 4.5 hours after symptom onset.”
At page 4 # paragraph 4 #, in Methods section
Comment 2. In the paragraph "Association of stroke risk scores...", in the newly added text, I got lost when considering whether NIHSS decreased with successful recanalization. Because if the median decrease in NIHSS is 0, I guess it's worse than if the median decrease in NIHSS is 5? Am I right? I also get the impression that the same text is duplicated and can be found again in the text below Figure 2.
Answer 2. We apologize for confusion after the revision of our manuscript. The formula for “change in NIHSS score after 24 hours” is as follows. “(Initial NIHSS score – NIHSS score at 24 hours = change in NIHSS score after 24 hours)” Therefore, as the reviewer noted, a median value of 0 for “change in NIHSS score after 24 hours” is worse than 5. The manuscript has been revised to avoid this confusion.
From
“Stroke severity was defined using the National Institutes of Health Stroke Scale (NIHSS) score and the neurologic change after EVT 24 hours was defined as a change in NIHSS score after 24 hours (Initial NIHSS score – NIHSS score at 24 hours = change in NIHSS score after 24 hours).”
To
“Stroke severity was defined using the National Institutes of Health Stroke Scale (NIHSS) score and the neurologic change after 24 hours of EVT was defined as the difference between the initial NIHSS score and the NIHSS score at 24 hours (Initial NIHSS score – NIHSS score at 24 hours = change in NIHSS score after 24 hours). Therefore, if this value was positive, it means neurological improvement, o means no improvement, and negative means neurological worsened.
At page 3 # paragraph 3 #, in Methods section
From
“(median 0 [IQR -2‒3] vs. median 5 [IQR 0‒10], p<0.001)”
To
“(Initial NIHSS score – NIHSS score at 24 hours, median 5 [IQR 0‒10] vs. median 0 [IQR -2‒3], p<0.001)”
At page 7 # paragraph 3 #, in Results section
From
“(median 0 [IQR -1‒2] vs. median 5 [IQR 1‒10], p<0.001)”
To
“(Initial NIHSS score – NIHSS score at 24 hours, median 5 [IQR 1‒10] vs. median 0 [IQR -1‒2], p<0.001)”
At page 10 # paragraph 6 #, in Results section
Comment 3. Please explain why diabetes was removed from Table 2?
Answer 3. Sorry for the confusion. DM was not used as a cofounder when multivariable analysis was performed from the original version of manuscript. Since DM was used as a variable for all stroke risk scores, multicollinearity is highly likely to occur when DM used as a cofounder of multivariable logistic analysis. This has already been mentioned in the statistical analyses section of the manuscript (original version).
“Body mass index (BMI) and p < 0.1 (excluding age and DM, which are common overlapping variables for all stroke risk scores) from univariable analysis were entered in multivariable analysis.”
At page 4 # paragraph 1 #, in Statistical analyses
However, although DM was deleted from Table S3 of Supplemental data, DM-related contents were not accidentally deleted from Table 2 (original version). Therefore, it has been removed from the revision version. Sorry for not letting you know this in advance.
Best Regards

This manuscript is a resubmission of an earlier submission. The following is a list of the peer review reports and author responses from that submission.
Round 1
Reviewer 1 Report
The authors assessed the association between the pre-admission CHADS2, CHA2DS2-VASc, ATRIA, and Essen scores and unsuccessful recanalization (mTICI ≤2a) after mechanical thrombectomy (MT) in a set of 501 patients consecutively included in the Korean nationwide multicenter registry Selection Criteria in Endovascular Thrombectomy and Thrombolytic therapy (SECRET). According to the results of the multivariate logistic regression analysis, an association was found between these stroke risk scores and unsuccessful recanalization – CHADS2 score: OR 1.551, 95% CI 1.198‒2.009, p=0.001; CHA2DS2-VASc score: OR 1.269, 95% CI 1.080‒1.492, p=0.004; ATRIA score: OR 1.089, 95% CI 1.011‒1.174, p=0.024; and Essen score: OR 1.469, 95% CI 1.167‒1.849, p=0.001. The CHADS2 score had the highest AUC value and differed significantly only from the Essen score. The authors concluded, that these stroke risk scores could be beneficial when predicting recanalization in stroke patients undergoing MT.
The whole paper is well written, with clear Methods and sound Results sections. The authors state correctly, that their results suggest that most of the factors related to unsuccessful recanalization in MT patients overlap with most factors related to the occurrence of stroke in patients with atrial fibrillation.
Major comment
- Patients eligible for MT according to the valid guidelines cannot be excluded from this therapy. This should be emphasized in the Discussion.
- The authors should also discuss, in which particular situations the prediction of the probability of recanalization based on the assessment of pre-admission stroke risk scores would be beneficial.
Minor comment
- What does the term “occlusion of atherosclerosis” used in the Introduction mean? Is it an “atherosclerotic occlusion” or “occlusion due to atherosclerosis”?
Author Response
Response to Reviewers’ comments
December 07, 2021
Manuscript ID: jcm-1427981
Title: Association between CHADS2, CHA2DS2-VASc, ATRIA, and Essen stroke risk scores and unsuccessful recanalization after endovascular thrombectomy in acute ischemic stroke patients
Dear reviewers & editor
First of all, we would like to thank the reviewers and editorial board members for their time and valuable comments on our manuscript. We have addressed our opinions on each comment from two reviewers in this response letter, and made several changes and corrections to our original manuscript.
We tried to revise our manuscript according to reviewer’s suggestions as much as possible, and revised parts are written in red color texts in the manuscript. We hope that the revisions in the manuscript and our accompanying responses will be sufficient to make our manuscript suitable for publication in Journal of Clinical Medicine.
We shall look forward to hearing from you at your earliest convenience.
Sincerely yours,
Tae-Jin Song, MD, PhD.
Department of Neurology, Seoul Hospital, Ewha Womans University College of Medicine, 260, Gonghang-daero, Gangseo-gu, Seoul, 07804, Republic of Korea
Tel: +82-2-2650-2677, Fax: +82-2-2650-5958; E-mail: knstar@ewha.ac.kr
Hyo Suk Nam, MD, PhD
Department of Neurology, Yonsei University College of Medicine
50-1 Yonsei-ro, Seodaemoon-gu, Seoul, 03722, Korea
Tel: 82-2-2228-1617, Fax: 82-2-393-0705; E-mail: hsnam@yuhs.ac
Please see the attachment

Reviewer 2 Report
Thank you very much for the opportunity to review this paper, which is of a very good standard and addresses the significant issue, from the practitioner's point of view, of selecting the group of patients who will benefit most from mechanical thrombectomy, or of detecting factors that influence the unfavourable outcome of TM.
However, I have a few comments.
Firstly, on the number of patients. From what I understood, 501 patients who had a TM procedure were selected from the database. So why do the patients of "IA thrombolysis alone" treatment appear in Table 1?
Secondly, the authors gave the average time from symptom onset to the performance of the various reperfusion procedures, but in the rest of the article, they do not refer to time as a critical factor modifying many parameters related to stroke treatment. Is it possible to perform additional statistics to look at time as a likely cofounder?
Thirdly, can the authors state what the intervals between rtPA and TM looked like in the groups that had good and bad recanalization?
Fourthly, I am also missing information on whether good recanalization has translated into any change in NIHSS, even after 24 hours? As a practitioner, I have seen some situations where good recanalization did not necessarily translate into a reduction in NIHSS, and some situations where incomplete recanalization resulted in significant clinical improvement. Perhaps the scales in question can predict the clinical outcome?
Regarding the tests - they all include diabetes as an essential component. Diabetes was also an important factor in differentiating the groups with good and bad reperfusion. In the discussion, I feel some shortfall regarding this finding. Maybe, we do not have to perform all those scales and only check the glucose level?
The Atria test also appears to be the most difficult to include in clinical practice, as it has in its measurements the results of a urinalysis and biochemical blood tests, which, as we know, in the interest of obtaining the shortest possible time to reperfusion, are severely limited in the acute phase. In their discussion, could the authors consider the importance of renal factors? Are there solid indications that they will have to be taken into account in the future?
Author Response

(The authors gave the same response as above.)
